# Dereplication, Annotation, and Characterization of 74 Potential Antimicrobial Metabolites from *Penicillium Sclerotiorum* Using t-SNE Molecular Networks

**DOI:** 10.3390/metabo11070444

**Published:** 2021-07-08

**Authors:** Téo Hebra, Nicolas Elie, Salomé Poyer, Elsa Van Elslande, David Touboul, Véronique Eparvier

**Affiliations:** CNRS, Institut de Chimie des Substances Naturelles, Université Paris-Saclay, UPR 2301, 91198 Gif-sur-Yvette, France; Teo.HEBRA@cnrs.fr (T.H.); Nicolas.ELIE@cnrs.fr (N.E.); salome.poyer@gmail.com (S.P.); elsa.van-elslande@cnrs.fr (E.V.E.)

**Keywords:** *Penicillium sclerotiorum*, azaphilone, molecular networks, dereplication

## Abstract

Microorganisms associated with termites are an original resource for identifying new chemical scaffolds or active metabolites. A molecular network was generated from a collection of strain extracts analyzed by liquid chromatography coupled to tandem high-resolution mass spectrometry, a molecular network was generated, and activities against the human pathogens methicillin-resistant *Staphylococcus aureus*, *Candida albicans* and *Trichophyton rubrum* were mapped, leading to the selection of a single active extract of *Penicillium sclerotiorum* SNB-CN111. This fungal species is known to produce azaphilones, a colorful family of polyketides with a wide range of biological activities and economic interests in the food industry. By exploring the molecular network data, it was shown that the chemical diversity related to the *P. sclerotiorum* metabolome largely exceeded the data already reported in the literature. According to the described fragmentation pathways of protonated azaphilones, the annotation of 74 azaphilones was proposed, including 49 never isolated or synthesized thus far. Our hypothesis was validated by the isolation and characterization of eight azaphilones, among which three new azaphilones were chlorogeumasnol (**63**), peniazaphilone E (**74**) and 7-deacetylisochromophilone VI (**80**).

## 1. Introduction

For decades, natural products have been the most productive source of leads for new drugs, including antimicrobials [1]. Nevertheless, new chemical scaffolds are always required to extend therapeutic arsenals in order to address global public health problems, such as antibiotic resistance. To this end, new ecological niches must be explored, and their relative chemodiversity must be evaluated [2].

Among the ecological niches that have been little studied, the microorganisms associated with insects are rising in interest. One million five thousand insect species (Arthropods) have been formally described to date [3]. Arthropods colonize almost all terrestrial habitats, including forests, deserts and coasts. These organisms are also colonized by microorganisms located in different compartments, such as cuticles, digestive systems and glands [4]. Insect–microorganism interactions have been widely studied within apocrites (bees, wasps, ants), but few studies are related to termite–microorganism interactions outside the trophobiosis [5,6,7,8]. However, examples of antimicrobial compounds produced by microorganisms associated with termites from French Guiana were previously published in the literature, especially by the research group of D. Stien and V. Eparvier [9,10,11,12].

In the present work, the termite-associated microorganism strain collection at the Institute of Chemistry of Natural Substances (ICSN) was reinvestigated using newly developed metabolomic tools.

Indeed, time-consuming purifications and rediscovery of active known compounds are recurring problems in natural product research. Dereplication using mass spectrometry-based molecular networking approaches can be implemented to overcome these drawbacks, allowing for the targeting of specific metabolites through the tandem mass spectrometry dataset organization that they confer [13,14,15,16,17,18].

After mapping the bioactivity against three human pathogens, i.e., methicillin-resistant *Staphylococcus aureus*, *Candida albicans* and *Trichophyton rubrum*, on the molecular network, the strain extract of *Penicillium sclerotiorum* SNB-CN111, with minimal inhibitory concentrations (MICs) of 16 and 32 µg.mL^−1^ against *Candida albicans* and *Trichophyton rubrum,* respectively, and without significant cytotoxicity (cell survival greater than 80% at a concentration of 10 µg.mL^−1^), was selected. *Penicillium sclerotiorum* is a fungus known to produce antimicrobial fungal polyketides called azaphilones, among which chlorinated compounds have been described [19,20,21]. Moreover, azaphilones attract high interest in industry, particularly for natural food coloring [22]. The purpose of this study was to annotate and characterize azaphilones in the SNB-111 extract strain by determining their fragmentation patterns by tandem mass spectrometry, exploring the related molecular network, proposing putative in silico structures and finally confirming this approach by isolating new bioactive compounds.

## 2. Results

### 2.1. Penicillium Sclerotiorum Selection

The 109 crude extracts of microorganisms associated with termites were first analyzed by liquid chromatography coupled to tandem high-resolution mass spectrometry (LC-HRMS/MS) in data-dependent analysis (DDA) mode and in positive ion mode. Extracts were also biologically tested against two human pathogens, i.e., *Candida albicans* and *Trichophyton rubrum* (Appendix A). The MS/MS data generated were processed to create a molecular network (MN) by MS/MS similarity related to chemical similarity using MetGem software (Appendix A) [13,14]. Briefly, MS/MS data were converted to vectors, and cosine scores between each vector were calculated to evaluate the distance of structural homology. The t-SNE algorithm implemented in MetGem software was employed to generate the final distance map, called the molecular network. The t-SNE allowed exploration of the dataset as a whole and preserved chemical proximity between clusters compared to the GNPS-like network. Metadata, such as the proposed chemical formula, retention times, relative areas of extracted ion peaks or activity of each extract, can be mapped on the MN using color mode. Once the molecular network was generated, the MS² libraries of standard molecules were queried, and known metabolites were annotated. Specific clusters related to *P. sclerotiorum* SNB-CN111 were enlightened by mapping strain identity on the network (Appendix A). Using MS/MS database search engines, five MS/MS spectra were annotated as azaphilone analogs: sclerotiorin (**1**), sclerotioramine (**2**), ochrephilone (**3**), isochromophilone I (**4**) and isochromophilone VI (**5**) (Appendix A).

### 2.2. Penicillium Sclerotiorum Metabolism Dereplication

To optimize the identification of minority compounds produced by *P. sclerotiorum*, a scale-up culture was performed. Bioguided fractionation was undertaken, leading to ten fractions whose antimicrobial activities were determined (Appendix A), leading to 5 active fractions against *Trichophyton rubrum* with MICs below 32 µg.mL^−1^. In parallel, a t-SNE molecular network including bioactivity levels against *Trichophyton rubrum* was generated from LC–MS/MS data of the ten *P. sclerotiorum* fractions (Figure 1). Before isolation, a first annotation by querying MS/MS spectral databases was achieved. To clarify our molecular network feature annotation and to respect standards for reproducible science, annotation levels proposed by the Metabolomics Standards Initiative (MSI) were used [23]. Level 0 corresponds to an unambiguous 3D structure obtained from isolated pure compounds; level 1 corresponds to identified compound by comparison with a standard; level 2 corresponds to putative annotation (e.g., MS/MS library comparison or tentative structure); level 3 includes a chemical class assignment; and level 4 means unknown molecules. Sclerotiorin (**1**), sclerotioramine (**2**), ochrephilone (**3**), isochromophilone I (**4**) and isochromophilone VI (**5**), as well as isochromophilone IX (**6**), hypocrellone A (**7**), atlantinone A (**8**), diketopiperazine (DKP: cyclo-PhePro (**9**)), fatty acids (pinolenic acid (**10**) and linolenic acid (**11**)) and contaminants (dioctyl phthalate (**12**), cyclopentasiloxane (**13**)), were finally annotated in silico (Appendix A). Several clusters of molecules in active fractions were thus annotated using MS/MS similarities (Figure 1).

Subsequently, a second step in the dereplication process was carried out. First, molecules described in the Atlas of Natural Products as compounds **1** to **7** analogs were searched. Then, a second search was carried out on the Reaxys database on compounds structurally similar to molecules isolated from natural resources [24]. Structures proposed through the literature review were annotated with level 3 using only exact mass and taxonomic information. Five molecules, i.e., geumsanol A-C and G (**14–17)** and eupeniazaphilone C (**18**), were therefore annotated in cluster C, where hypocrellone (**7**) was first dereplicated [25,26,27]. Two molecules, i.e., isochromophilone IX (**6**) or penazaphilone F (**19**) and penazaphilone D (**20**), were also annotated in cluster B, including compounds **2** and **5** [28,29]. Isochromophilone IV (**21**) and sclerketide B isomer (**22**) completed the annotation of cluster A, including sclerotiorin (**1**) [30,31]. Finally, 5-chloroisorotiorin (**23**) completed cluster D, featuring molecules **3** and **4** [32]. Using the literature review, it was thus possible to annotate 10 more azaphilones produced by *P. sclerotiorum* SNB-CN111 (Appendix A). Despite these two first in silico steps, more than approximately one hundred molecules related to azaphilones remain unannotated.

### 2.3. *In Silico* Azaphilone Structure Prediction Using MS/MS Data and a t-SNE Molecular Network

According to the first in silico dereplication steps, 4 distinct subclasses of azaphilone produced by the *P. sclerotiorum* SNB-CN111 strain were uncovered. In fact, these azaphilone subclasses presented one or several modifications of the same scaffold, including acylations on R_1_, a lactone ring at R_1_–R_2_, the presence of an oxygen or a functionalized nitrogen on Y, a diol or a methylethylene at R_3_–R_4_ and finally chlorine or hydrogen at position X. (Figure 2). Isolated or synthetic azaphilones do not display all this combinatorial diversity in the literature [21]. To further annotate the azaphilone-related metabolome from the *P. sclerotiorum* SNB-CN111 strain, all available chemical information, including MS/MS data, chemical formulas, biosynthetic pathways and literature surveys, was gathered and combined. Five distinct clusters were analyzed in more detail (Figure 3, Figure 4 and Figure 5).

First, cluster A, containing annotated sclerotiorin (**1**), isochromophilone IV (**21**) and sclerketide B isomer (**22**), was clearly separated from others on the t-SNE molecular network. Two nonchlorinated analogs, **24** and **25,** of these three compounds were first annotated according to exact mass measurements below 5 ppm, isotopic patterns, MS/MS data and high cosine score values of 0.93 and 0.81 for pairs **1**/**24** and **22**/**25**, respectively. In particular, three common neutral losses of 42.0106, 60.0211 and 88.0160 Da, related to the losses of C_2_H_2_O, C_2_H_4_O_2,_ and C_3_H_4_O_3_, were observed in the spectra of acetylated azaphilones **1** and **24**. Similarly, propionated compounds **22** and **25** led to losses of 56.0262, 74.0368 and 102.0317 Da, corresponding to the neutral losses of C_3_H_4_O, C_3_H_6_O_2_ and C_4_H_6_O_3_, respectively. This observation, supported by the literature evidence for the fragmentation of protonated esters [33,34,35,36], suggests that the nature of the acyl group can be identified from this typical fragmentation pattern, as described in Figure 4.

According to this first rule of azaphilone fragmentation, compounds **26** and **27** were annotated as sclerotiorin analogs with butanoyl groups at R_1_ (Appendix A), whereas compounds **28** and **29** were annotated as sclerotiorin analogs with pentanoyl groups (Appendix A). Protonated nonacetylated compounds **30** and **31** were also detected (Appendix A). For these two particular compounds, fragments at *m*/*z* 181.0051 were observed for chlorinated molecule **30** and at *m*/*z* 147.0441 for hydrogenated molecule **31** (Appendix A). Dechlorosclerotiorin (**32**), dechlorobenzoylsclerotiorin (**33**), aminobenzoylsclerotiorin (**34**), 5-epoxysclerotiorin (**35**), benzoylsclerotiorin (**36**) and methoxysclerotiorin (**37)** were similarly annotated in cluster A. Finally, compound **38** was annotated as an analog of molecule **26** with hydroxylation on the butanoyl moiety (Appendix A).

Using the same methodology, subclusters B1 and B2, initially containing molecules **2** and **5**, respectively, were further annotated with (Figure 4) compounds **39–46** in cluster B1 (Figure 5) and compounds **47–49** in cluster B2 (Appendix A). These two subclusters were close in t-SNE MN, allowing us to deduce that they exhibit strong structural similarities. Because nitrogenated azaphilones spontaneously transform from their oxygenated analogs, molecule **42** can be annotated by comparison with compound **22** [21]. The two molecules had a cosine score of 0.61 and shared eight neutral losses. It was also observed in cluster B2 that compound **5** is an analog of compound **2** with an ethanol function on its nitrogen. Moreover, nitrogen-substituted azaphilones **5**, **6**, **19** and **20** were annotated on the northern part of t-SNE MN. Thus, cluster B2 only contained azaphilones **50** to **60** with substituted nitrogen (Appendix A). Molecules **61** and **62** in cluster B3, initially containing molecule **6**, were finally annotated, taking into account the same hypothesis (Appendix A).

Cluster C, containing hypocrellone A (**7**), was then annotated. As previously described in the other clusters, a chlorinated analog of **7** was observed (**63**) (Figure 6 and Appendix A). Two common neutral losses of 86.0732 (C_5_H_10_O) and 132.0786 (C_6_H_12_O_3_) were detected for compounds **7**, **14–17** and **63**. Compound **64** and its chlorinated analog **65** also exhibited the neutral loss of 132.0786 Da (Appendix A), leading to the formation of intense fragments at *m*/*z* 259.1329 and 293.0939, respectively. The same fragments were found for compounds **15** and **63** formed from the C_6_H_12_O_3_ moiety and the loss of CO_2_ from their lactone rings. Analogs with one less unsaturated **66** and nitrogen analogs **67** and **68** were also observed (Appendix A).

In cluster D, containing compound **3** analogs, molecules **70** and **71** bear epoxyde groups on alkyl chains, indicating that they are key intermediates between molecules **3**/**4** and **15**/**24** (Appendix A) [37,38]. Compounds **72–75** were annotated as unchlorinated azaphilone analogs exhibiting a lactone ring (Appendix A).

Finally, **76** and **77**, already isolated in a previous study, were identified in cluster E, as well as **78** and **79**, epoxide intermediates between **76**–**77** and **64**–**65** (Appendix A) [39].

### 2.4. Isolation and Characterization of Compounds

To confirm the in silico annotation, azaphilones from the most active fractions on *T. rubrum* (Appendix A) were isolated for complete structural elucidation. Annotated and known compounds **1**, **2**, **5**, **23** and **75** or newly annotated compounds **63**, **74** and **80** were thus purified and structurally characterized (Figure 7). The four known compounds were identified by ^1^H and ^13^C NMR and data comparison with the literature (Appendix A, Appendix A) [40]. Additional HSQC, COSY and HMBC experiments completed the characterization of **63**, **74** and **80** (Appendix A).

Compound **63** was obtained as a yellow oil, and its molecular formula was determined to be C_23_H_27_ClO_7_ based on the ESI–HRMS experiment ([M + H]^+^ peak at *m*/*z* 451.1523 calcd for C_23_H_27_ClO_7_H^+^, 451.1518, err. −1.1 ppm). The similarity of NMR data with compound **3** (Appendix A) indicated the presence of a lactone ring and a chlorine atom on carbon 5. Thus, ^1^H NMR and ^13^C spectroscopic data of **63** were analyzed and compared with the literature [25,41]. The azaphilone scaffold was identified by HMBC correlations of H1/C-3, C-4a, C-5 and C-8, H4/C-3, C-5, C-8, C-8a and H18/C-6, C-7, C-8, downfield chemical shifts of C-1 (δ_C_ 146.5) and C-3 (δ_C_ 157.3) and chemical shifts from ketocarbonyl carbon C-6 (δ_C_ 184.5). The side chain was connected to C-3 by the HMBC correlation of H-9 and H-10. The lactone moiety was confirmed by the presence of four additional carbon resonances comprising two carbonyls (δ_C_ 199.7 and 168.1), one methine (δ_C_ 57.3), one methyl group (δ_C_ 30.3), a COSY correlation between H-8 and H-3″ and HMBC from H-3″/C-2″. Ketocarbonyl carbon C-4″ and C-5″ were connected to C-3″ by HMBC correlation of H3″/C4″ and H5″/C3″, C4″. The chlorine atom was positioned on C-5 because it had no HSQC correlation. Finally, a typical correlation of transcoupled olefinic protons was observed in COSY with correlations between H-9/H-10, H-17/H-12/H-13/H-16 and H-13/H-14/H-15 (43). Observation of two carbons, C-11 (δ_C_ 75.9 ppm) and C-12 (δ_C_ 78.4 ppm), displayed hydroxylated carbon chemical shifts and permitted the establishment of the side chain as 3,5-dimethylhept-1-ene-3,4-diol (Appendix A, S117, Appendix A). This attribution is in accordance with the reported NMR characterization of **15**, an analog of **63** without chlorine at position X [25]; this compound was named chlorogeumasnol.

Molecule **74** was isolated as a purple amorphous oil, and its molecular formula was determined to be C_23_H_26_NO_4_ based on ESI–HRMS data ([M + H]^+^ at *m*/*z* 380.1863, calcd for C_23_H_25_NO_4_H+, 380.1856, err. −1.8 ppm). The 3,5-dimethyl-1,3-heptadienyl unit, the azaphilone scaffold and their connection were identified as described for **63**. The lactone ring was established with the HMBC correlation of H-5″ with C-4″, C-3″ and C-2″ and by comparison with its analog described in the literature [28,32] (Appendix A). Compound **74** was named peniazaphilone E.

Compound **80** exhibited a high peak intensity in LC–MS and was close to cluster D in t-SNE MN (Appendix A). By studying MS/MS fragmentation, compound **80** was expected to be an analog of **5** with a hydroxyl group at R_1_ (Appendix A). Compound **80** was obtained as a red oil, and its molecular formula was determined to be C_21_H_27_ClNO_4_ based on the ESI–HRMS experiment ([M + H]^+^ peak at *m*/*z* 392.1604 calcd for C_21_H_26_ClNO_4_H+, 392.1623, err. 4.9 ppm). The 3,5-dimethyl-1,3-heptadienyl unit, the azaphilone scaffold and their connection were identified as described for **63**. The ethanol chain was established by the COSY correlation between H1′ and H2′, as well as the C1′ (δ_C_ 56.6) and C2′ (δ_C_ 60.5) chemical shifts and HMBC correlation of H-1/C-1′ (Appendix A, Appendix A). Molecule **80** was named 7-deacetylisochromophilone VI.

Crystal structures for molecules **1** and **5** were obtained, allowing us to determine the absolute configuration of each chiral carbon atom (Appendix A, Appendix A). Thus, the absolute configuration of C-7 of all other isolated compounds was determined by comparison of the circular dichroism of each isolated molecule with compounds **1** and **5**.

All isolated compounds were tested on two fungal human pathogens and showed moderate MICs (Table 1). Molecules **1**, **2** and 2**3** displayed the best activities against *T. rubrum*, with MICs of 32, 64 and 32 µg.mL^−1^, respectively. Only compound **23** showed anticandidal activity, with an MIC of 64 µg.mL^−1^. In the literature, an enantiomer on position 7 of rubrotiorin (**23**) is reported with an IC_50_ of 0.6 µg.mL^−1^ on *Candida albicans* [42].

## 3. Discussion

*P. sclerotiorum* SNB-CN111 specialized metabolism was deeply examined among 109 extracts of microorganisms associated with termites from French Guiana. This strain showed both antimicrobial biological activity and a wide variety of molecules from the azaphilone class.

The LC–MS/MS and in silico dereplication process provided an example of how an extended analysis of MS/MS data can lead to large-scale azaphilone annotation. First, it could be demonstrated that the fractionation of the crude extract allowed a better dereplication of the azaphilones. The number of features increased from 382 to 2953, leading to a more complete MN using t-SNE visualization. However, only a few azaphilones were dereplicated by querying the MS/MS. To enlarge our annotation, a literature survey was performed. Height azaphilones were annotated after this second dereplication step coming from the isolation process (**14–21**). Eight additionally reported compounds not detected by the first dereplication round were then identified thanks to our in silico strategy: **24**, **34**, **35**, **40**, **72** and **75–77** [32,39].

In silico strategies were tested, but the results were not satisfactory enough for azaphilone analogs [43,44,45,46]. Indeed, in silico strategies display 17–25% to 87–93% annotation accuracy for “known–unknown” metabolites, depending on the database boost [46]. However, these strategies do not perform correctly for “unknown–unknown” metabolites derived from enzymatic and chemical transformation or for original structures, as was the case for azaphilones extracted from *P. sclerotiorum* SNB-CN111. At present, there is a need to increase our ability to accurately annotate “unknown–unknown” characteristics to identify compounds that cannot be isolated for various reasons.

The primary annotation based on HRMS data permitted us to differentiate analogs with and without chlorine atoms on the carbon at position 5 due to a typical difference of 33.9610 combined with a change in isotopic pattern related to the characteristic abundance of ^37^Cl [47,48]. Interestingly, only two unchlorinated *N*-azaphilones (**41** and **53**) were observed. It was supposed that chlorine atoms contribute to azaphilone scaffold affinity for primary amines by electroattractive effects. The nature of the acylation on the hydroxyl at position 7 was simply identified according to typical neural losses, as described in Figure 4. In this way, four different acylations of azaphilone, i.e., acetylation, propionylation, butanoylation and pentanoylation, were systemically associated with three different azaphilone scaffolds (**1**/**22**/**26**/**28**, **2**/**42**/**45**/**44**, **24**/**25**/**27**/**29**). As expected when performing reverse-phase LC, the CH_2_ increment linearly increased the retention time of each molecule (Appendix A). More unique modifications, such as benzoyl and hydroxylation of the acyl moiety, were also identified. Moreover, for azaphilones bearing a benzoylation (**33**, **34**, **36**), the benzoyl moiety leads to the formation of the main ion fragment, whereas the azaphilone scaffold constitutes the neutral loss (Appendix A, S39). In contrast, for azaphilone with linear acylation, the acyl moiety is the main neutral loss, and azaphilone is the main ion fragment. This particular fragmentation pathway due to the presence or absence of a labile proton at position α of the carbonyl was previously described for mitorubrin azaphilone [35].

One of the specificities of azaphilone chemistry is its capacity for spontaneous conversion of oxygen atoms into nitrogen groups at position 2. Thus, a mass shift of 0.9843 between the compounds is directly related to the spontaneous exchange of O by NH. Neutral losses of 71.9844 corresponding to the loss of a CO_2_ group from the lactone ring and CO from the azaphilone scaffold were then observed (**3**, **4**, **15**, **63**, **75**, **76**). Furthermore, four pairs of azaphilones and their diol analogs were found (**3**/**15**, **4**/**63**, **64**/**76** and **66**/**77**), indicating a monooxygenase-epoxide hydrolase pathway. Due to a mass shift of 15.9956, the presence of an epoxide on two particular azaphilones was hypothesized (**70**, **71**). Four trios of azaphilone, including epoxy-azaphilone and diol-azaphilone, were finally annotated (**3**/**70**/**15, 4**/**71**/**63**, **76/78/64** and **77**/**79**/**65**). The azaphilones with a lactone in R_1_ and R_2_ were positioned in cluster D, and those with diols in R_3_ and R_4_ were positioned in cluster C. Overall, five combinatorial modifications of azaphilone were identified. Thanks to the study of their specific fragmentation patterns by MS/MS, the biosynthesis pathway of azaphilones was proposed in line with the literature (Figure 8) [21,37,38,47,48]. All annotations were confirmed and validated by the isolation of compounds **1**, **2**, **5**, **23**, **63**, **74** and **75** identified in the molecular network.

The highest activities related to fractions F6, F7 and F8 were linked to features depicted in clusters A, B1, B3 and D. Many of the azaphilones grouped in these clusters were minor compounds, and only **1**, **2**, **5**, **23**, **63**, **74**, **75** and **80** were isolated and structurally characterized. Experimental MIC measurements using these pure compounds confirmed moderate activities against human pathogens. It can be assumed either that the most active molecule(s) have not been isolated thus far due to low abundance in the fractions or that azaphilones have synergistic activities.

Another property of azaphilones is their strong absorbance, which results in yellow, orange, red or violet molecules. For example, compound **1** shows a maximum absorbance at 361 nm, **5** at 370 nm and **23** at 356 nm [7,30,49]. When using acetonitrile as the solvent, molecules **74** and **75** also exhibit two maximal absorbances at 424/548 nm and 428/541 nm, respectively.

## 4. Materials and Methods

### 4.1. General Experimental Procedures

Optical rotations were measured at 20 °C in acetonitrile using an Anton Paar MCP 300 polarimeter in a 100-mm-long 350 μL cell. UV spectra were recorded at 20 °C in acetonitrile or methanol using a PerkinElmer Lambda 5 spectrophotometer. Electronic circular dichroism spectra were acquired at 20 °C in acetonitrile on a JASCO J-810 spectropolarimeter. NMR spectra were recorded on Bruker 300, 500, 600 and 700 MHz spectrometers (Bruker, Rheinstetten, Germany). The chemical shifts (δ) are reported as ppm based on the solvent signal, and coupling constants (*J*) are in hertz. Preparative HPLC was conducted with a Gilson system equipped with a 322 pumping device, a GX-271 fraction collector, a 171 diode array detector and a prep ELSII. All solvents were HPLC grade, purchased from Sigma-Aldrich (Saint-Quentin-Fallavier, France).

### 4.2. Isolation and Identification of Termite Mutualistic Microorganisms

#### 4.2.1. General Identification Procedure

The taxonomic marker analyses were externally performed by BACTUP, France. The identification of the fungi was conducted by amplification of the ITS4 or ITS1 region of ribosomal DNA and the bacterial isolates were identified on the basis of 16S rDNA sequence analysis. The sequences were aligned with DNA sequences from GenBank, NCBI (http://www.ncbi.nlm.nih.gov, accessed on 7 June 2021), using BLASTN 2.2.28. The sequences were deposited in the GenBank for accession numbers.

#### 4.2.2. Isolation and Identification of Penicillium Sclerotiorum SNB-CN111

The strain was isolated from a *Nasutitermes similis* termite aerial nest sampled in Piste de Saint-Elie (N 05° 01,838′ W 052° 44,606′) in French Guiana. The strain SNB-CN111 from the strain library collection at ICSN was identified as *Penicillium sclerotiorum*. A sample submitted for amplification and nuclear ribosomal internal transcribed spacer region ITS4 sequencing allowed for strain identification by NCBI sequence comparison. The sequence has been registered in the NCBI GenBank database (http://www.ncbi.nlm.nih.gov, accessed on 4 September 2013) under registry number KJ023726. 

### 4.3. Culture and Extraction of Microorganisms

#### 4.3.1. General Cultivation and Extraction Procedure

All strains, including bacteria, were cultivated on solid PDA medium at 26 °C for 15 days, on 3 Petri dishes of 14 cm diameter (150 cm^2^). On a large scale, the microorganisms were cultivated under the identical conditions with 130 Petri dishes of 14 cm diameter (2 m^2^). The contents of the Petri dishes were transferred into a large container and macerated with EtOAc for 24 h. The organic solvent was collected by filtration under vacuum, washed with water in a separating funnel and evaporated to dryness under reduced pressure.

#### 4.3.2. Extraction of SNB-CN111

*P. sclerotiorum* was cultivated on 330 Petri dishes (14 cm diameter) at 28 °C for 15 days on potato dextrose agar (PDA) medium (Dominique Dutscher SAS, Brumath, France). The culture medium containing the mycelium was cut into small pieces and macerated three times at room temperature with ethyl acetate (EtOAc) on a rotary shaker (70 rpm) for 24 h. The contents were extracted with 10 L of EtOAc using a separatory funnel. Insoluble residues were removed via filtration and the organic phase was washed three times with an equivalent volume of water (H_2_O), dried with anhydrous solid Na_2_SO_4_ and then evaporated using a rotary evaporator under reduced pressure and temperature of 30 °C to yield a crude extract (6.5 g).

### 4.4. Isolation of Compounds

The whole crude extract was fractionated by reversed-phase flash chromatography (Grace Reveleris, Grace, MD, USA) using a 120 g C18 column and ultraviolet (UV) and evaporative light scattering detectors (ELSDs). A linear gradient of H_2_O/formic acid (99.9/0.1) (A)–acetonitrile/formic acid (99.9/0.1) (B) (from 5% B to 100% B in 40 min, flow rate at 80 mL · min^−1^) followed by a second gradient of acetonitrile/formic acid (99.9/0.1) (B)–methylene chloride (C) (from 50% C to 100% C in 15 min, flow rate at 80 mL · min^−1^) was performed to generate 11 fractions labeled F1 to F11. F5 (810 mg) was refractionated by reversed-phase flash chromatography using a 40 g C18 column and ultraviolet (UV) and evaporative light scattering detector (ELSD). A linear gradient of H_2_O/formic acid (99.9/0.1) (A)–acetonitrile/formic acid (99.9/0.1) (B) (60% B for 10 min, 60% B to 65% B for 6 min, 65% B to 75% B for 4 min then 100% B for 10 min, flow rate at 80 mL · min^−1^) followed by a second gradient of acetonitrile/formic acid (99.9/0.1) (B)–methylene chloride (C) (50% C to 100% C in 15 min, flow rate at 80 mL · min^−1^) was performed to generate 9 fractions labeled F5F1 to F5F9.

Fractions of interest (F5F2, F5F3, F5F4, F5F5, F6 and F7) were submitted to preparative HPLC. Further fractionation of F5F2 (H_2_O/formic acid (99.9/0.1) (A)–acetonitrile/formic acid (99.9/0.1) (B) gradient, 48% B for 5 min, 48% B to 55% B in 15 min and 100% B for 5 min) led to the isolation of the new compound **63** (1.2 mg, *t_R_* = 11.1 min), peniazaphilone A (**75**, 1.8 mg, *t_R_* = 13.6 min) and isochromophilone VI (**5**, 30 mg, *t_R_* = 16.0 min). Further fractionation of F5F3 (H_2_O/formic acid (99.9/0.1) (A)–acetonitrile/formic acid (99.9/0.1) (B) gradient, 48% B for 5 min, 48% B to 55% B in 15 min and 100% B for 5 min) led to the isolation of the new compound **74** (2.2 mg, *t_R_* = 17.0 min). Further fractionation of F5F4 (H_2_O/formic acid (99.9/0.1) (A)–acetonitrile/formic acid (99.9/0.1) (B) gradient, 50% B for 30 min and 100% B for 5 min) led to the isolation of sclerotioramine (**2**, 8 mg, *t_R_* = 15.0). Further fractionation of F5F5 (H_2_O/formic acid (99.9/0.1) (A)–acetonitrile/formic acid (99.9/0.1) (B) gradient 60% B to 75% B in 30 min and 100% B for 5 min) led to the isolation of new compound **80** (0.56 mg, *t_R_* = 5.3 min). Further fractionation of F6 (H2O/formic acid (99.9/0.1) (A)–acetonitrile/formic acid (99.9/0.1) (B) gradient, 65% B in 30 min) led to the isolation of sclerotiorin (**1**, 6 mg, *t_R_* = 15.2 min). Further fractionation of F7 (H2O/formic acid (99.9/0.1) (A)–acetonitrile/formic acid (99.9/0.1) (B) gradient, 70% B to 80% B in 20 min, 80% B to 85% B in 5 min and 100% B for 5 min) led to the isolation of 5-chloroisorotiorin (**23**, 13.8 mg, *t_R_* = 21.0 gradient, 70% B to 80% B in 20 min).

Sclerotiorin (**1**): Yellow amorphous oil or yellow needles; [α]^20^_D_ 200 (*c* 0.1 g.L^−1^, ACN), UV (ACN) λmax (ε), 286 nm (4 100 Lmol^−1^.cm^−1^), 361 nm (9 800 Lmol^−1^.cm^−1^) UV (meOH) λmax (ε), 271 nm (16 000 Lmol^−1^.cm^−1^), 385 nm (21 300 Lmol^−1^.cm^−1^), ^1^H NMR (600 MHz, CDCl_3_) δ_H_ 7.90 (1H, s, H-1), 6.61 (1H, s, H-4), 6.51 (1H, d, J = 15.9 Hz, H-9), 7.03 (1H, d, J = 15.9 Hz, H-10), 5.67 (1H, d, J = 10.0 Hz, H-12), 2.445(1H, sept, J = 7.0 Hz, H-13), 1.39 (1H, sept, J = 7.0 Hz, H-14a), 1.29 (1H, sept, J = 7.0 Hz, H-14b), 0.83 (3H, t, J = 7.5 Hz, H-15), 0.98 (3H, d, J = 6.8 Hz, H-16), 1.81 (3H, s, H-17), 1.53 (3H, s, H-18), 2.13 (3H, s, H-20), 13C NMR (600 MHz, CDCl3) δC 152.8 (CH, C-1), 158.4 (C, C-3), 110.8 (CH, C-4), 138.8 (C, C-4a), 106.6 (C, -5), 186.0 (C, C-6), 84.7 (C, C-7), 192.0 (C, C-8), 114.8 (C, C-8a), 115.9 (CH, C-9), 143.0 (CH, C-10), 132.2 (C, C-11), 149.0 (CH, C-12), 35.3 (CH, C-13), 30.2 (CH2, C-14), 12.1 (CH3, C-15), 20.2 (CH3, C-16), 12.5 (CH3, C-17), 22.7 (CH3, C-18), 170.3(C, C-19), 20.3 (CH3, C-20), ESI–HRMS *m*/*z* [M + H]^+^ 391.1324 (calcd for C_21_H_23_ClO_5_H^+^, 391.1307, err. −4.4 ppm).

Sclerotioramine (**2**): Red amorphous oil; [α]^20^_D_ 300 (*c* 0.1 g.L^−1^, ACN), UV (ACN) λmax (ε), 343 nm (16 400 Lmol^−1^.cm^−1^), UV (MeOH) λmax (ε), 331 nm (20,000 Lmol^−1^.cm^−1^), 488 nm (1900 Lmol^−1^.cm^−1^), ^1^H NMR (500 MHz, CDCl_3_) δ_H_ 7.93 (1H, s, H-1), 6.86 (1H, s, H-4), 6.13 (1H, d, J = 16.1 Hz, H-9), 7.04 (1H, d, J = 16.7 Hz, H-10), 5.69 (1H, d, J = 9.9 Hz, H-12), 2.47 (1H, sept, J = 7.0 Hz, H-13), 1.40 (1H, sept, J = 7.0 Hz, H-14a), 1.30 (1H, sept, J = 7.0 Hz, H-14b), 0.84 (3H, t, J = 7.5 Hz, H-15), 0.99 (3H, d, J = 6.8 Hz, H-16), 1.83 (3H, s, H-17), 1.57 (3H, s, H-18), 2.16 (3H, s, H-20), ^13^C NMR (500 MHz, CDCl_3_) δ_C_ 138.4 (CH, C-1), 146.3 (C, C-3), 110.3 (CH, C-4), 147.1 (C, C-4a), 101.5 (C, -5), 183.7 (C, C-6), 85.4 (C, C-7), 193.3 (C, C-8), 114.2 (C, C-8a), 116.4 (CH, C-9), 142.9 (CH, C-10), 132.0 (C, C-11), 148.7 (CH, C-12), 35.1 (CH, C-13), 30.6 (CH_2_, C-14), 12.0 (CH_3_, C-15), 20.1 (CH_3_, C-16), 12.4 (CH_3_, C-17), 23.6 (CH_3_, C-18), 170.9 (C, C-19), 20.6 (CH_3_, C-20), ESI–HRMS *m*/*z* [M + H]^+^ 390.1460 (calcd for C_21_H_24_ClNO_4_H^+^, 390.1467, err. 1.7 ppm).

Isochromophilone VI (**5**): Red amorphous oil or red needles; [α]^20^_D_ 440 (*c* 0.1 g.L^−1^, ACN), UV (ACN) λmax (ε), 371 nm (9 700 Lmol^−1^.cm^−1^), UV (MeOH) λmax (ε), 369 nm (13 200 Lmol^−1^.cm^−1^), 480 nm (2 200 Lmol^−1^.cm^−1^), ^1^H NMR (500 MHz, CDCl_3_) δH 7.83 (1H, s, H-1), 7.00 (1H, s, H-4), 6.23 (1H, d, J = 15.0 Hz, H-9), 6.91 (1H, d, J = 15.8 Hz, H-10), 5.68 (1H, d, J = 9.2 Hz, H-12), 2.46 (1H, m, H-13), 1.44 (1H, sept, J = 7.2 Hz, H-14a), 1.32 (1H, sept, J = 7.2 Hz, H-14b), 0.86 (3H, t, J = 7.5 Hz, H-15), 1.00 (3H, d, J = 6.8 Hz, H-16), 1.83 (3H, s, H-17), 1.53 (3H, s, H-18), 2.14 (3H, s, H-20), 3.99 (2H, m, H-1′), 3.91 (2H, m, H-2′), ^13^C NMR (500 MHz, CDCl_3_) δ_C_ 142.0 (CH, C-1), 144.8 (C, C-3), 111.8 (CH, C-4), 148.5 (C, C-4a), 102.2 (C, -5), 184.4 (C, C-6), 84.9 (C, C-7), 194.0 (C, C-8), 114.6 (C, C-8a), 115.1 (CH, C-9), 145.1 (CH, C-10), 131.7 (C, C-11), 148.0 (CH, C-12), 35.0 (CH, C-13), 30.0 (CH_2_, C-14), 12.0 (CH3, C-15), 20.2 (CH_3_, C-16), 12.6 (CH_3_, C-17), 23.3 (CH_3_, C-18), 170.2 (C, C-19), 20.3 (CH_3_, C-20), 55.4 (CH_2_, C-1′), 60.9 (CH_2_, C-2′), ESI–HRMS *m*/*z* [M + H]^+^ 434.1731 (calcd for C_23_H_28_ClNO_5_H^+^, 434.1729, err. −0.5 ppm).

5-chloroisorotiorin (**23**), orange amorphous oil; [α]^20^_D_ 200 (*c* 0.1 g.L^−1^, ACN), UV (ACN) λmax (ε), 361 nm (8 800 Lmol^−1^.cm^−1^), UV (MeOH) λmax (ε), 373 nm (14 700Lmol^−1^.cm^−1^), 558 nm (1 700 Lmol^−1^.cm^−1^), ^1^H NMR (500 MHz, MeOD) δH 8.88 (1H, s, H-1), 6.81 (1H, s, H-4), 6.35 (1H, d, J = 17.1 Hz, H-9), 7.20 (1H, d, J = 17.2 Hz, H-10), 5.77 (1H, d, J = 9.7 Hz, H-12), 2.52 (1H, m, H-13), 1.46 (1H, m, H-14a), 1.35 (1H, m, H-14b), 0.89 (3H, t, J = 7.5 Hz, H-15), 1.03 (3H, d, J = 6.5 Hz, H-16), 1.89 (3H, s, H-17), 1.68 (3H, s, H-18), 2.57 (3H, s, H-5″), ESI–HRMS *m*/*z* [M + H]^+^ 415.1300 (calcd for C_23_H_23_ClO_5_H^+^, 415.1307, err. 1.6 ppm).

Chlorogeumsanol B (**63**), yellow amorphous oil; [α]^20^_D_ -160 (*c* 0.1 g.L^−1^, ACN), UV (ACN) λmax (ε), 367 nm (8 200 Lmol^−1^.cm^−1^), UV (MeOH) λmax (ε), 292 nm (5 000 Lmol^−1^.cm^−1^), 386 nm (14 400 Lmol^−1^.cm^−1^), 1H NMR (500 MHz, CDCl3) δH 7.42 (1H, s, H-1), 6.56 (1H, s, H-4), 3.84 (1H, d, J = 12.5 Hz, 1H), 6.40 (1H, d, J = 15.5 Hz, H-9), 6.65 (1H, d, J = 15.7 Hz, H-10), 3.49 (1H, d, J = 1.2 Hz, H-12), 1.7 (1H, m, H-13), 1.39 (1H, sept, J = 7.4 Hz, H-14a), 1.31 (1H, m, H-14b), 0.90 (3H, t, J = 7.4 Hz, H-15), 0.95 (3H, d, J = 6.7 Hz, H-16), 1.33 (3H, s, H-17), 1.59 (3H, s, H-18), 3.76 (1H, d, J = 12.5 Hz, H-3″), 2.45 (3H, s, H-5″) δC 146.5 (CH, C-1), 157.3 (C, C-3), 106.1 (CH, C-4), 140.4 (C, C-4a), 113.6 (C, -5), 184.5 (C, C-6), 83.1 (C, C-7), 42.6 (CH, C-8), 110.2 (C, C-8a), 120.2 (CH, C-9), 145.5 (CH, C-10), 75.9 (C, C-11), 78.4 (CH, C-12), 35.5 (CH, C-13), 28.7(CH_2_, C-14), 12.0 (CH_3_, C-15), 24.0 (CH_3_, C-16), 13.5 (CH_3_, C-17), 23.4 (CH_3_, C-18), 168.1 (C, C-2″), 57.3 (CH, C-3″), 199.7 (C, C-4″), 30.3 (CH_3_, C-5″), ESI–HRMS *m*/*z* [M + H]^+^ 451.1523 (calcd for C_23_H_27_ClO_7_H^+^, 451.1518, err. -1.1 ppm).

Peniazaphilone E (**74**) purple amorphous oil; [α]^20^_D_ 330 (*c* 0.1 g.L^−1^, ACN), UV (ACN) λmax (ε), 335 nm (12 700 Lmol^−1^.cm^−1^), 548 nm (9 100 Lmol^−1^.cm^−1^), 424 nm (8 500 Lmol^−1^.cm^−1^), UV (MeOH) λmax (ε), 331 nm (2 400 Lmol^−1^.cm^−1^), ^1^H NMR (500 MHz, CDCl3) δH 9.32 (1H, s, H-1), 6.80 (1H, s, H-4), 6.78 (1H, s, H-5), 6.28 (1H, d, J = 16.4 Hz, H-9), 7.57 (1H, d, J = 16.8 Hz, H-10), 5.82 (1H, d, J = 10.4 Hz, H-12), 2.49 (1H, m, H-13), 1.43 (1H, m, H-14a), 1.34 (1H, m, H-14b), 0.86 (3H, t, J = 7.4 Hz, H-15), 0.99 (3H, d, J = 6.7 Hz, H-16), 1.88 (3H, s, H-17), 1.80 (3H, s, H-18), 2.49 (3H, s, H-5″); ^13^C NMR (300 MHz, CDCl_3_) δC 141.3 (CH, C-1), 148.8 (C, C-3), 117.1 (CH, C-4), 154.0 (C, C-4a), 99.1 (CH, C,-5), 195.5 (C, C-6), 86.9 (C, C-7), 172.5 (C, C-8), 117.5 (C, C-8a), 116.5 (CH, C-9), 144.6 (CH, C-10), 132.5 (C, C-11), 149.7 (CH, C-12), 35.4 (CH, C-13), 30.3 (CH_2_, C-14), 12.2 (CH_3_, C-15), 20.5 (CH_3_, C-16), 12.4 (CH_3_, C-17), 29.9 (CH_3_, C-18), 175.0 (C, C-2″), 101.2 (CH, C-3″), 193.5 (C, C-4″), 28.6 (CH_3_, C-5″), ESI–HRMS *m*/*z* [M + H]^+^ 380.1863 (calcd for C_23_H_25_NO_4_H^+^, 380.1856, err. −1.8 ppm).

Peniazaphilone A (**75**), purple amorphous oil; [α]^20^_D_ 600 (*c* 0.1 g.L^−1^, ACN), UV (ACN) λmax (ε), 338 nm (14 200 Lmol^−1^.cm^−1^), 428 nm (12 400 Lmol^−1^.cm^−1^), 541 nm (10 900 Lmol^−1^.cm^−1^), UV (MeOH) λmax (ε), 342 nm (6 400 Lmol^−1^.cm^−1^), ^1^H NMR (500 MHz, CDCl3) δH 7.92 (1H, s, H-1), 6.77 (1H, s, H-4), 6.29 (1H, d, J = 16.2 Hz, H-9), 6.95 (1H, d, J = 15.8 Hz, H-10), 5.70 (1H, d, J = 9.8 Hz, H-12), 2.46 (1H, m, H-13), 1.43 (1H, sept, H-14a), 1.33 (1H, sept, H-14b), 0.86 (3H, t, J = 7.5 Hz, H-15), 1.00 (3H, d, J = 6.6 Hz, H-16), 1.86 (3H, s, H-17), 1.57 (3H, s, H-18), 4.13 (2H, m, H-1′), 4.06 (2H, m, H-2′), 2.36 (3H, s, H-5″), ^13^C NMR (500 MHz, CDCl3) δ_C_ 142.6 (CH, C-1), 149.8 (C, C-3), 117.0 (CH, C-4), 150.3 (C, C-4a), 98.0 (CH, -5), 194.4 (C, C-6), 85.3 (C, C-7), 171.5 (C, C-8), 117.2 (C, C-8a), 114.8 (CH, C-9), 146.2 (CH, C-10), 132.0 (C, C-11), 149.0 (CH, C-12), 35.1 (CH, C-13), 28.6 (CH_2_, C-14), 12.0 (CH_3_, C-15), 20.2 (CH_3_, C-16), 12.6 (CH_3_, C-17), 30.2 (CH_3_, C-18), 55.4 (CH_2_, C-1′), 60.9 (CH_2_, C-2′), 172.3 (C, C-2″), 103.6 (C, C-3″), 193.9 (C, C-4″), 30.3 (CH_3_, C-5″), ESI–HRMS *m*/*z* [M + H]^+^ 424.2134 (calcd for C_25_H_29_NO_5_H^+^, 424.2118, err. −3.7 ppm).

7-deacetylisochromophilone VI (**80**), amorphous red oil; [α]^20^_D_ 60 (*c* 0.1 g.L^−1^, ACN), UV (ACN) λmax (ε), 365 nm (3 200 Lmol^−1^.cm^−1^), UV (MeOH) λmax (ε), 361 nm (3 300 Lmol^−1^.cm^−1^), ^1^H NMR (500 MHz, DMF-d7) δH 8.10 (1H, s, H-1), 7.01 (1H, s, H-4), 6.7 (1H, d, J = 15.8 Hz, H-9), 7.23 (1H, d, J = 15.7 Hz, H-10), 5.87 (1H, d, J = 9.5 Hz, H-12), 2.53 (1H, m, H-13), 1.43 (1H, m, H-14a), 1.32 (1H, m, H-14b), 0.87 (3H, t, J = 7.4 Hz, H-15), 1.01 (3H, d, J = 6.6 Hz, H-16), 1.92 (3H, s, H-17), 1.41 (3H, s, H-18), 4.38 (2H, m, H-1′), 3.88 (2H, m, H-2′); ^13^C NMR (700 MHz, DMF-d7) δC 142.6 (CH, C-1), 149.8 (C, C-3), 109.7 (CH, C-4), 145.9 (C, C-4a), 98.8 (C, -5), 187.1 (C, C-6), 83.1 (C, C-7), 197.5 (C, C-8), 115.1 (C, C-8a), 117.2 (CH, C-9), 144.3 (CH, C-10), 133.2 (C, C-11), 146.8 (CH, C-12), 34.8 (CH, C-13), 29.8 (CH_2_, C-14), 11.8 (CH_3_, C-15), 20.1 (CH_3_, C-16), 12.3 (CH_3_, C-17), 28.2 (CH_3_, C-18), 56.6 (CH_2_, C-1′), 60.5 (CH_2_, C-2′), ESI–HRMS *m*/*z* [M + H]^+^ 392.1604 (calcd for C_21_H_26_ClNO_4_H^+^, 392.1623, err. 4.9 ppm).

### 4.5. LC–MS/MS Analysis

Crude extracts of all SNB-CN strains, cultivated on PDA and extracted as previously described for *Penicillium sclerotiorum* SNB-CN111 together with fractions from *Penicillium sclerotiorum* SNB-CN111, were prepared at 1 mg.mL^−1^ in methanol and filtered on a 0.45 µm PTFE membrane. LC–MS/MS experiments were performed with a 1260 Prime HPLC (Agilent Technologies, Waldbronn, Germany) coupled with an Agilent 6540 Q-ToF (Agilent Technologies, Waldbronn, Germany) tandem mass spectrometer. LC separation was achieved with an Accucore RP-MS column (100 × 2.1 mm, 2.6 μm, Thermo Scientific, Les Ulis, France) with a mobile phase consisting of H_2_O/formic acid (99.9/0.1) (A)–acetonitrile/formic acid (99.9/0.1) (B). The column oven was set at 45 °C. Compounds were eluted at a flow rate of 0.4 mL · min^−1^ with a gradient from 5% B to 100% B in 20 min and then 100% B for 3 min. The injection volume was fixed at 5 μL for all analyses. For electrospray ionization source, mass spectra were recorded in positive ion mode with the following parameters: gas temperature 325 °C, drying gas flow rate 10 L.min^−1^, nebulizer pressure 30 psi, sheath gas temperature 350 °C, sheath gas flow rate 10 L.min^−1^, capillary voltage 3500 V, nozzle voltage 500 V, fragmentor voltage 130 V, skimmer voltage 45 V, Octopole 1 RF Voltage 750 V. For ESI, internal calibration was achieved with two calibrants, purine and hexakis (1 h,1 h,3 h-tetrafluoropropoxy) phosphazene (*m*/*z* 121.0509 and *m*/*z* 922.0098), providing a high mass accuracy better than 3 ppm. The data-dependent MS/MS events were acquired for the five most intense ions detected by full-scan MS, from the 200–1000 *m*/*z* range, above an absolute threshold of 1000 counts. Selected precursor ions were fragmented at a fixed collision energy of 30 eV and with an isolation window of 1.3 amu. The mass range of the precursor and fragment ions was set as *m*/*z* 200–1000.

Isolated compounds from *Penicillium sclerotiorum* SNB-CN111 fractions were prepared at 0.1 mg.mL^−1^ in methanol and filtered on a 0.45 µm PTFE membrane. The isolated compounds were analyzed according to the same procedure.

### 4.6. Data Processing and Analysis

The data files were converted from the d standard data format (Agilent Technologies) to mzXML format using MSConvert software, part of the ProteoWizard package 3.0 (21). All mzxml values were processed using MZmine2v51 as previously described [16]. Mass detection was realized with an MS1 noise level of 1000 and an MS/MS noise level of 0. The ADAP chromatogram builder was employed with a minimum group size of scans of 3, a group intensity threshold of 1000, a minimum highest intensity of 1000 and *m*/*z* tolerance of 0.008 (or 20 ppm). Deconvolution was performed with the ADAP wavelet algorithm according to the following settings: S/N threshold = 10, minimum feature height = 1000, coefficient/area threshold = 10, peak duration range 0.01–1.5 min and *t_R_* wavelet range 0.00–0.04 min. MS/MS scans were paired using a *m*/*z* tolerance range of 0.05 Da and *t_R_* tolerance range of 0.5 min. Isotopologs were grouped using the isotopic peak grouper algorithm with a *m*/*z* tolerance of 0.008 (or 20 ppm) and a *t_R_* tolerance of 0.2 min. Peaks were filtered using a feature list row filter, keeping only peaks with MS/MS scans (GNPS). Adduct identification, i.e., sodium- or potassium-cationized species, was performed on the peak list with a retention time tolerance of 0.1 min, a *m*/*z* tolerance of 0.008 or 20 ppm and a maximum relative peak height of 150%. A complex search, such as dimers, was performed with a retention time tolerance of 0.1 min, a *m*/*z* tolerance of 0.008 or 20 ppm and a maximum relative peak height of 150%. Peak alignment was performed using the join aligner with a *m*/*z* tolerance of 0.008 (or 20 ppm), a weight for *m*/*z* at 20, a retention time tolerance of 0.2 min and weight for *t_R_* at 50. The MGF file and the metadata were generated using the export/submit to GNPS option.

Molecular networks were calculated and visualized using MetGem 1.3 software [14], and MS/MS spectra were window-filtered by choosing only the top 6 peaks in the ±50 Da window throughout the spectrum. The data were filtered by removing all peaks in the ± 17 Da range around the precursor *m*/*z*. The m/z tolerance windows used to find the matching peaks were set to 0.02 Da, and cosine scores were kept in consideration for spectra sharing at least 2 matching peaks. The number of iterations, perplexity, learning rate and early exaggeration parameters were set to 5000, 25, 200 and 12 for the t-SNE view.

Figures were generated using R and related packages (ggplot2, Rcolorbrewer and gridextra), MetGem export function and ChemDraw Professional 16.0 (PerkinElmer). NMR spectra were processed and analyzed using TopSpin 3.6.2 (Bruker, Rheinstetten, Germany).

### 4.7. X-ray Structure Determination of Compounds 1 and 5

Crystals of compounds **1** and **5** were obtained by slow evaporation at 4 °C. A suitable crystal was selected for each of them, mounted on a nylon loop and fixed with oil. Then, X-ray diffraction and crystallographic data were collected at room temperature using redundant ω scans on a Rigaku XtaLabPro single-crystal diffractometer using microfocus Mo Kα radiation and an HPAD PILATUS3 R 200K detector.

Using Olex2 [50], the structures were readily solved by intrinsic phasing methods (SHELXT [51]) and by full-matrix least-squares methods on F2 using SHELXL [52]. The nonhydrogen atoms were refined anisotropically, and most of the hydrogen atoms were identified in difference maps and were treated as riding on their parent atoms.

For each structure, the Flack parameter [53] was refined. The determination of the absolute structure was confirmed by using Bayesian statistics on Bijvoet differences [54] based on the Olex2 results.

All the molecular graphics presented here were computed with Mercury 2020.3.0 [55].

Crystallographic data for the two structures (1 and 5) have been deposited in the Cambridge Crystallographic Data Centre database (the deposition numbers are CCDC 2085749 and CCDC 2085750, respectively). Copies of the data can be obtained free of charge from the CCDC at www.ccdc.cam.ac.uk (accessed on 7 July 2021).

### 4.8. Biological Assays

The crude extracts and pure isolated compounds were tested on the human pathogenic microorganisms *Candida albicans* (ATCC 10213), methicillin-resistant *Staphylococcus aureus* (ATCC 33591) and *Trichophyton rubrum* (SNB-TR1). The test was performed in conformance with reference protocols from the European Committee on Antimicrobial Susceptibility Testing [56]. The minimal inhibitory concentration value was obtained after 48 h for *C. albicans*, 24 h for MRSA and 72 h for *T. rubrum*. Vancomycin (for bacteria) and itraconazole (for fungi) were used as positive controls.

For cytotoxic assays, the crude extracts and isolated compounds were tested in triplicate at concentrations of 10 µg.mL^−1^ and 1 µg.mL^−1^ in the MRC5 cell line (ATCC CCL-171, Human Lung Fibroblast Cells), following the procedure described by Tempête et al. [57].

## 5. Conclusions

We annotated 74 azaphilone analogs, including 49 new molecules. Among them, eight azaphilones were isolated, including three new azaphilones. The structural data are in agreement with the structure predictions from MS/MS data organized in molecular networks. According to the newly established fragmentation pathways, azaphilones can now be efficiently dereplicated in silico in complex mixtures, even if a low amount of samples is available. Large collections of *Penicilium* can now be screened for the production of undescribed azaphilones, allowing us to better understand their biosynthetic pathways.

This article represents the first study showing the correct and robust annotation, by Moleculare Network methodology, of more than 70 structurally similar azaphilones in plant extracts. This result is due, in part, to the similar structure of the azaphilone-like compounds, but also to the highly specific fragmentation pathways during MS2 experiments. It should also be noted that the annotation was facilitated by the fact that many analogs had already been described in the literature.

## Figures and Tables

**Figure 1 metabolites-11-00444-f001:**
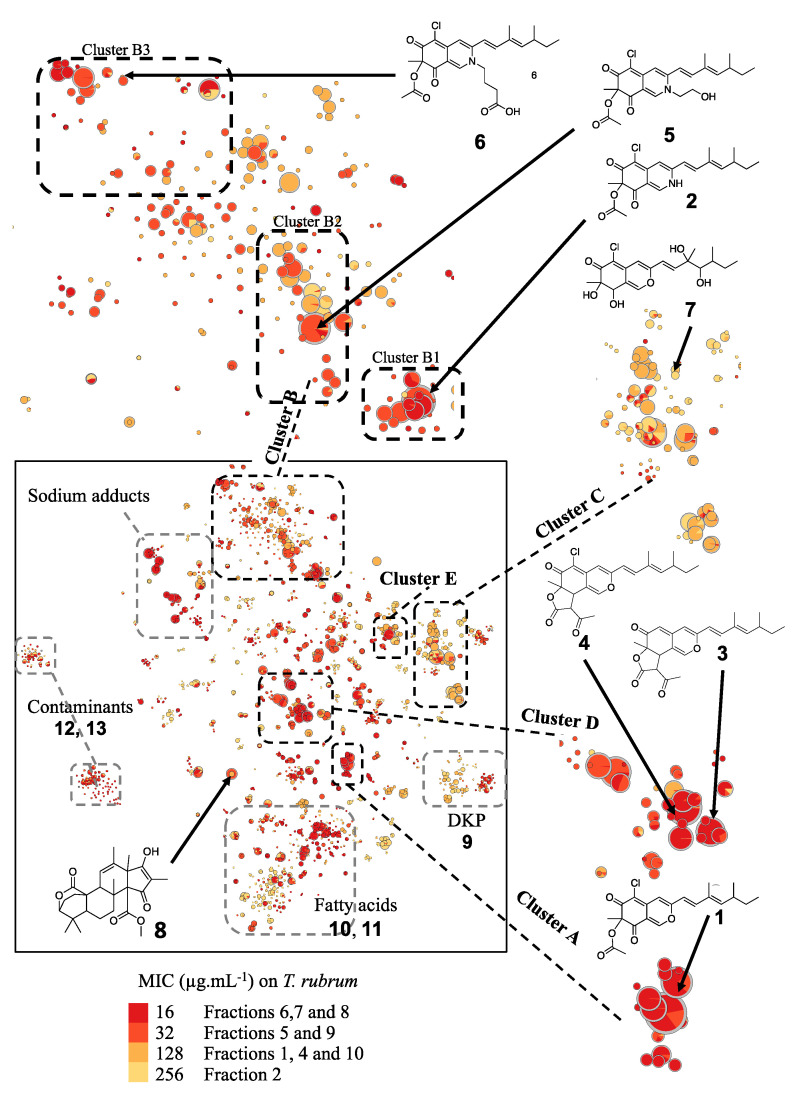
t-SNE molecular network representation constructed on MS² homology, with the size of the nodes related to their intensity and color to minimal inhibitory concentration (MIC) of fractions against *T. rubrum* with dereplicated features from MS² standard database query. Azaphilone derivative clusters are boxed in black.

**Figure 2 metabolites-11-00444-f002:**
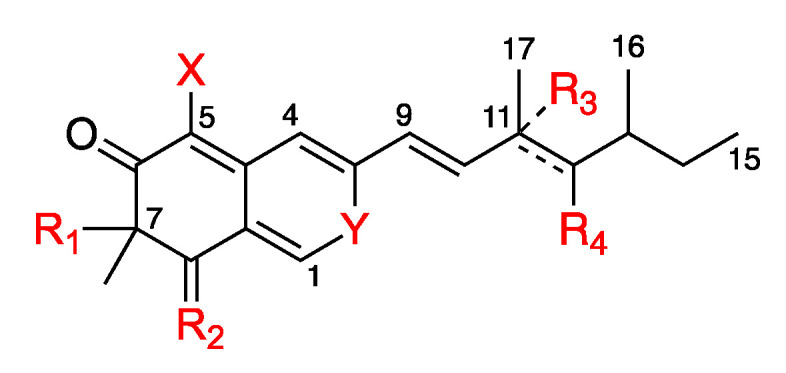
R_1_ can possess acylation of different sizes or form a lactone ring with R_2_. Y can be O, NH or N-alkylation, R_3_ and R_4_ can form an alcene until C-11 and C-12 or an epoxyde or are hydroxyles, and X can form either a hydrogen or a chlorine atom. These modifications can be combined.

**Figure 3 metabolites-11-00444-f003:**
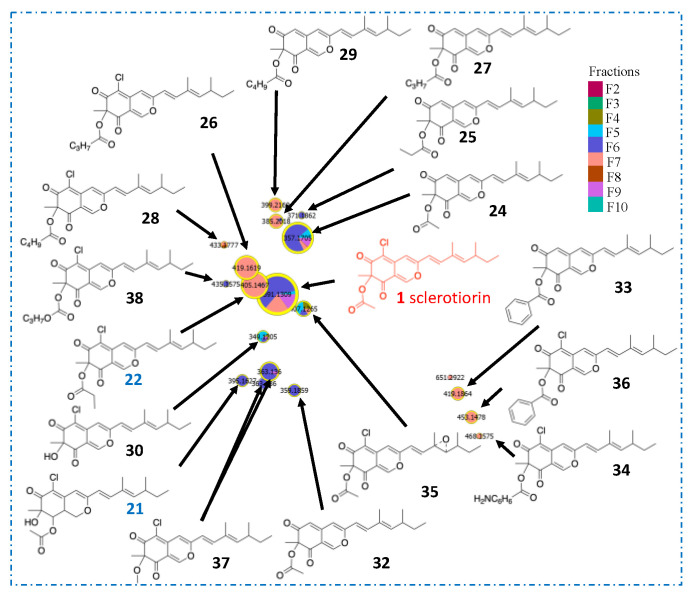
Annotation cluster A included sclerotiorin (**1**) with hypothetical structures proposed from molecular network data. In red and blue are molecules identified directly in MN (isochromophilone IV (**21**) and sclerketide B isomer (**22**)), in black are molecules annotated with MS/MS.

**Figure 4 metabolites-11-00444-f004:**
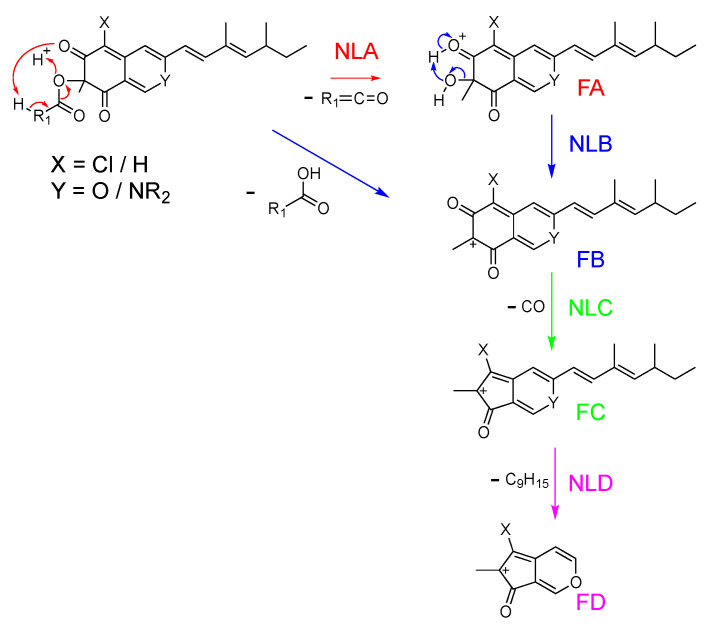
Proposed fragmentation pathway of molecules in clusters A and B. Depending on R_1_, X and Y elements. NL = neutral loss, F = fragmentation (below).

**Figure 5 metabolites-11-00444-f005:**
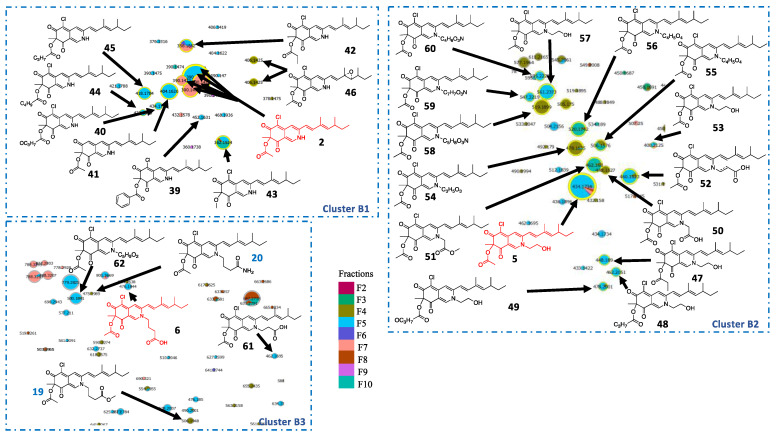
Annotation of cluster B with hypothetical structures proposed from molecular network data. In red and blue are molecules identified directly in MN (sclerotioramine (**2**), isochromophilone VI (**5**), isochromophilone IX (**6**), penazaphilone F (**19**) and penazaphilone D (**20**)), in black are molecules annotated with MS/MS.

**Figure 6 metabolites-11-00444-f006:**
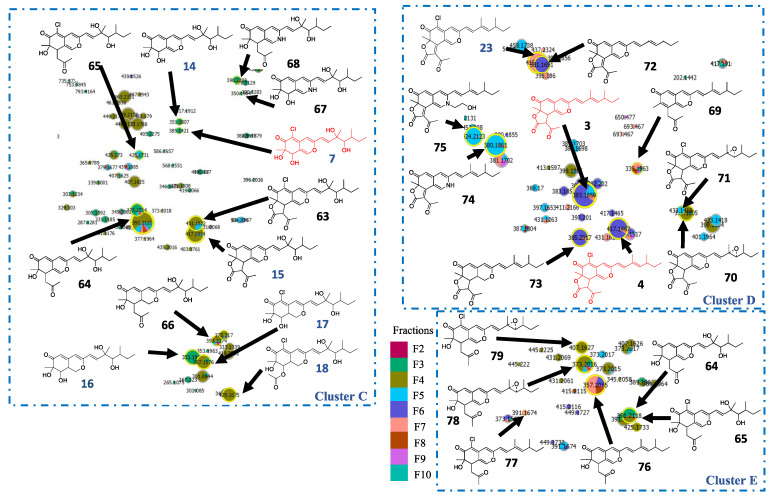
Annotation of clusters C, D and E with hypothetical structures proposed from molecular network data. In red and blue are molecules (ochrephilone (**3**), isochromophilone I (**4**) hypocrellone A (**7**), geumsanol A-C and G (**14–17)** and eupeniazaphilone C (**18**), 5-chloroisorotiorin (**23**)) identified directly in MN, in black are molecules annotated with MS/MS.

**Figure 7 metabolites-11-00444-f007:**
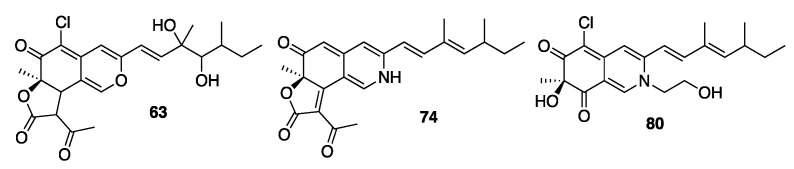
New azaphilone compounds isolated from CN111 extract.

**Figure 8 metabolites-11-00444-f008:**
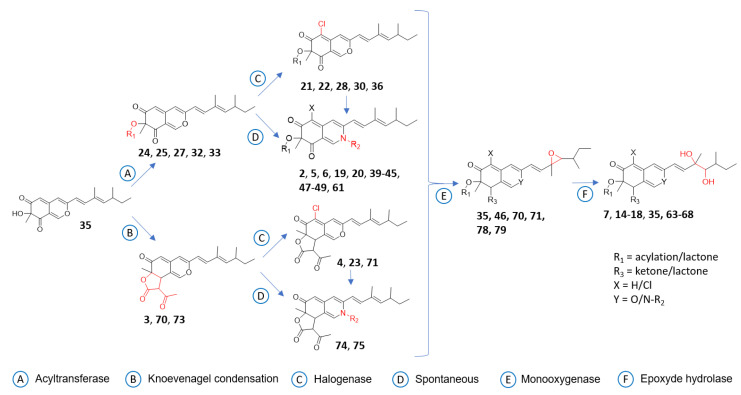
Summary of azaphilone modifications proposed as a metabolic pathway.

**Table 1 metabolites-11-00444-t001:** Minimal inhibitory concentrations in µg.mL^−1^ of isolated compounds against *C. albicans* and *T. rubrum*. Fluconazol was used as positive control.

Compounds/Fractions	*C. Albicans*	*T. Rubrum*
**1**	>64	32
**2**	>64	64
**5**	>64	>64
**23**	64	32
**63**	>64	>64
**74**	>64	>64
**75**	>64	>64
**80**	>64	>64
**F5**	256	32
**F6**	128	16
**F7**	128	16
**Control**	1	4

## Data Availability

The data presented in this study are available in supplementary materials.

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
