# Peer review of "Dereplication, Annotation, and Characterization of 74 Potential Antimicrobial Metabolites from Penicillium Sclerotiorum Using t-SNE Molecular Networks"

_metabolites, 2021, doi:10.3390/metabo11070444_

Round 1

Reviewer 1 Report

This manuscript report the annotation and characterization of 74 potential antimicrobial metabolites form a termites-associated bacteria, namely Penicillium sclerotium. Selected strain culture was extracted and analyzed by tandem HRMS, and the data obtained, together with antimicrobial activities data against some human resistant bacteria and fungi, were analyzed to generate a molecular network, according to their predicted chemical structure and reported biological activity. This system permitted the  dereplication and annotation of a large amount of secondary metabolites (74), among these, 8 compound were isolated and fully structural characterized by 1H,13CNMR and HRMS, and three new compound were discovered. The approach is quite innovative and allow the authors to identify various known and new natural products, belonging to the class of chromophilones or azaphilones. The manuscript is well organized and well written. In my opinion, even if the general presentation is not bad, it could be improved, because the vast amount of compounds presented in this work and the classification according to the clusters they belong to, which is not so clear and simple for the reader, which can get lost. Anyway, the results are well presented and the data are robust, in general the scientific soudness of this paper is high. Conclusions could be improved, giving some more details on the methodologies used for the achievement of annotation of so many compounds. Moreover, the system of molecular network resulted to be robust and to able to predict correctly the chemical structure of compounds that have been isolated successively. But what about the robustness in predicting the other structures? Same general percentages of accuracy of similar systems are reported, what about the one described herein?

You can find below some specific suggestions:

lines 117-118 two compounds have the same numeration (22): isochromophilone IV and 5-chloroisorotiorin, correct the numeration, since any compounds should have a single number;

In the Fig. 5 caption will be useful to insert the name of new isolated natural products;

line 299 acyl moiety instead of acylation moiety;

In all the experimental part some unit of measurement are reported with a dot which is not correct (e.g. 9 800 L.mol-1.cm-1 is not correct, it should be written Lmol-1cm-1);

line 534 and 535 ad a space between the.d and to.mzXML

Author Response

Dear Editor, dear reviewers,

We would like to thank all the reviewers for their comments on our manuscript. We implemented all the corrections and complete information’s requested. We hope that these corrections and answers will give you satisfaction.

You will find below our answers.

All the corrections are marked in blue.

Best regards

Véronique Eparvier

Reviewer 1

This manuscript report the annotation and characterization of 74 potential antimicrobial metabolites form a termites-associated bacteria, namely Penicillium sclerotium. Selected strain culture was extracted and analyzed by tandem HRMS, and the data obtained, together with antimicrobial activities data against some human resistant bacteria and fungi, were analyzed to generate a molecular network, according to their predicted chemical structure and reported biological activity. This system permitted the  dereplication and annotation of a large amount of secondary metabolites (74), among these, 8 compound were isolated and fully structural characterized by 1H,13CNMR and HRMS, and three new compound were discovered. The approach is quite innovative and allow the authors to identify various known and new natural products, belonging to the class of chromophilones or azaphilones. The manuscript is well organized and well written. In my opinion, even if the general presentation is not bad, it could be improved, because the vast amount of compounds presented in this work and the classification according to the clusters they belong to, which is not so clear and simple for the reader, which can get lost. Anyway, the results are well presented and the data are robust, in general the scientific soudness of this paper is high. Conclusions could be improved, giving some more details on the methodologies used for the achievement of annotation of so many compounds. Moreover, the system of molecular network resulted to be robust and to able to predict correctly the chemical structure of compounds that have been isolated successively. But what about the robustness in predicting the other structures? Same general percentages of accuracy of similar systems are reported, what about the one described herein?

In order to improve the readingness of the article, we included the name of the identified molecules by MN in the legend of Figures 3, 4 and 5.

This study showed that the use of molecular networks was an asset for compound annotation. In our case, this methodology allowed us to robustly account for the diversity of azaphilones in a highly complex mixture. This result was reached because azaphilones are structurally very diverse but the fragmentation pathways are very specific. It should also be noted that the annotation was facilitated by the fact that many analogues were already described in the literature. A sentence was added to the article.

You can find below some specific suggestions:

lines 117-118 two compounds have the same numeration (22): isochromophilone IV and 5-chloroisorotiorin, correct the numeration, since any compounds should have a single number;

Thank you for this remark. This is a mistake, the 5-chloroisorotiorin corresponds to number 23 and not 21. We corrected this point in the manuscript.

In the Fig. 5 caption will be useful to insert the name of new isolated natural products;

Done

line 299 acyl moiety instead of acylation moiety;

In all the experimental part some unit of measurement are reported with a dot which is not correct (e.g. 9 800 L.mol-1.cm-1 is not correct, it should be written Lmol-1cm-1);

Done

line 534 and 535 ad a space between the.d and to.mzXML

Done

Reviewer 2 Report

The work report on the chemical profile of P. sclerotiorum by MS analysis organized in moelcular networks, able to identify a large number of azaphilone molecules and supported by the isolation and structural characterization of some metabolites, The draback of this work is represented by the very poor biological activities, but the other results here obtained are relevant in allowing next dereplication approaches.

Author Response

(The authors gave the same response as above.)

Reviewer 3 Report

In this manuscript, the authors screened a termite-associated microbial collection for antifungal activity. One Penicillium strain of interest was further characterised as a producer of azaphilone natural products. Using MS fragmentation patterns and network analysis, a large number of azaphilones were tentatively identified. From the most active fractions, eight azaphilones were purified and their structures fully elucidated. This includes three new azaphilones reported here for the first time.

Overall, the manuscript is clearly written and a good example of the power of metabolite network analysis for natural product identification and dereplication.

Author Response

(The authors gave the same response as above.)
